# Synthesis and Cytotoxic Activity of 1,2,4-Triazolo-Linked *Bis*-Indolyl Conjugates as Dual Inhibitors of Tankyrase and PI3K

**DOI:** 10.3390/molecules27217642

**Published:** 2022-11-07

**Authors:** Prasanna A. Yakkala, Samir R. Panda, Syed Shafi, V. G. M. Naidu, M. Shahar Yar, Philemon N. Ubanako, Samson A. Adeyemi, Pradeep Kumar, Yahya E. Choonara, Eugene V. Radchenko, Vladimir A. Palyulin, Ahmed Kamal

**Affiliations:** 1Department of Pharmaceutical Chemistry, School of Pharmaceutical Education and Research, Jamia Hamdard, New Delhi 110062, India; 2Department of Pharmacology and Toxicology, National Institute of Pharmaceutical Education and Research (NIPER)-Guwahati, Guwahati 781101, India; 3Department of Chemistry, School of Chemical and Life Sciences, Jamia Hamdard, New Delhi 110062, India; 4Wits Advanced Drug Delivery Platform Research Unit, Department of Pharmacy and Pharmacology, School of Therapeutic Sciences, Faculty of Health Sciences, University of the Witwatersrand, Johannesburg 2193, South Africa; 5Department of Chemistry, Lomonosov Moscow State University, 119991 Moscow, Russia; 6Department of Pharmacy, Birla Institute of Technology & Science, Pilani, Hyderabad Campus, Hyderabad 500078, India

**Keywords:** 1,2,4-triazolo *bis*-indolyl conjugates, cytotoxicity, colon cancer, β-catenin pathway, PI3K/tankyrase inhibitors

## Abstract

A series of new 1,2,4-triazolo-linked *bis*-indolyl conjugates (**15a–r**) were prepared by multistep synthesis and evaluated for their cytotoxic activity against various human cancer cell lines. It was observed that they were more susceptible to colon and breast cancer cells. Conjugates **15o** (IC_50_ = 2.04 μM) and **15r** (IC_50_ = 0.85 μM) illustrated promising cytotoxicity compared to 5-fluorouracil (5-FU, IC_50_ = 5.31 μM) against the HT-29 cell line. Interestingly, **15o** and **15r** induced cell cycle arrest at the G_0_/G_1_ phase and disrupted the mitochondrial membrane potential. Moreover, these conjugates led to apoptosis in HT-29 at 2 μM and 1 μM, respectively, and also enhanced the total ROS production as well as the mitochondrial-generated ROS. Immunofluorescence and Western blot assays revealed that these conjugates reduced the expression levels of the PI3K-P85, β-catenin, TAB-182, β-actin, AXIN-2, and NF-κB markers that are involved in the β-catenin pathway of colorectal cancer. The results of the in silico docking studies of **15r** and **15o** further support their dual inhibitory behaviour against PI3K and tankyrase. Interestingly, the conjugates have adequate ADME-toxicity parameters based on the calculated results of the molecular dynamic simulations, as we found that these inhibitors (**15r**) influenced the conformational flexibility of the 4OA7 and 3L54 proteins.

## 1. Introduction

Colorectal cancer (CRC) is the third leading malignancy and one of the major causes of cancer-related mortality in developed as well as developing countries [1,2]. Around the globe, nearly 1.4 million cases and about 694,000 deaths have been reported every year [3,4]. The early diagnosis of gene mutations may help to prevent disease progression [5].

Twenty-four mutated genes, including PI3K, AKT, mTOR, KRAS, APC, GSK3β, TANKS, etc. [6,7], are mainly responsible for colorectal cancer. Among them, PI3KCA and glycogen synthase kinase 3β (GSK3β) are the key elements of the β-catenin destruction complex associated with major cancer cases [8,9]. The PI3K/Akt and Wnt/β-catenin pathways were found to be responsible for the majority of colon cancers [10].

The cross-talk between two signalling pathways, PI3K/Akt and Wnt/β-catenin, is associated with GSK3β [11], which consists of axin, adenomatous polyposis coli (APC), and casein kinase 1 [12]. GSK3β phosphorylates the β-catenin, which subsequently undergoes proteasomal degradation in the absence of Wnt [13]. Meanwhile, in the presence of Wnt, the activated Frizzled receptor associated with Dishevelled (DVL) intracellular signalling molecules stabilises β-catenin in the cytosol by disrupting the assembly of the β-catenin destruction complex [14].

Several natural and synthetic compounds bearing indoles [15,16], 1,2,4-triazoles [17], pyridines [18], quinazolines [19], *N*-amino-naphthalimide [20], and scaffolds (Figure 1) have demonstrated the inhibition of either PI3K or Wnt/β-catenin (tankyrase) pathways individually for their anti-cancer activity [11,21]. Compounds **1** and **2**, with indole nuclei, were found to deregulate multiple cellular signalling pathways, including the PI3K, AKT, and mTOR signalling pathways [16,22]. Further, the 1,2,4-triazole scaffold is a potent tankyrase inhibitor that downregulates auxin in the destructive complex, with promising cytotoxic properties (**3–6**) [17,23]. Particularly, 2-pyridyl-1,2,4-triazole scaffolds (**3** and **4**) have demonstrated promising tankyrase 1/2 inhibitory activity through regulating the Wnt signalling pathway. Compound G007-LK (**5**), bearing 1,2,4-triazole moiety, has demonstrated high selectivity towards tankyrase 1/2 and has also shown an excellent pharmacokinetic profile in mice, while WIKI4 (**6**) exerts its effects on the Wnt/β-catenin signalling pathway by inhibiting tankyrase 2 [8].

On the other hand, NVP-BKM120 (**7**), a quinazoline derivative is a promising PI3K/p-Akt inhibitor, as it decreases the cellular level of PI3K kinase [24]. Very few compounds, such as macrolactin A, were found to act as dual inhibitors of both the tankyrase and PI3K/Akt pathways [25] via reducing the nuclear β-catenin levels (tankyrase) and TCF/LEF transcriptional activity (in-vitro). The mouse model studies proved the dual inhibitory effect of SMA against tankyrase and PI3K/Akt [26]. Solberg and co-workers investigated the single and combined effects of the highly specific tankyrase 1/2 inhibitor G007-LK and the pan-class I (PI3K) inhibitor BKM120, and the results suggested that the combination of these two molecules successfully works as a dual inhibitor [11].

It is evident from the literature that there is a lack of PI3K and tankyrase dual inhibitors, and thus, there has been immense interest in the development of such dual inhibitors. Therefore, in view of the biological effects of 1,2,4-triazole and indole moieties for the inhibition of the tankyrase and PI3K/Akt pathways, some new 1,2,4-triazolo-linked *bis*-indolyl conjugates were synthesised and evaluated for their cytotoxic potential. Interestingly, these conjugates exhibited dual inhibiting activity for both tankyrase and PI3K. The rational design of the target molecules is depicted in Figure 2.

## 2. Results and Discussion

### 2.1. Chemistry

The 1,2,4-triazolo-linked *bis*-indolyl conjugates (**15a–p**) were prepared in a multistep synthetic approach, starting from ethyl 1H-indole-3-carboxylate (**9**), as shown in Figure 1.

Ethyl 1H-indole-3-carboxylate (**9**) was treated with hydrazine hydrate to obtain 1H-indole-3-carbohydrazide [27] (**10**). This compound was further reacted with different aliphatic and aromatic isothiocyanates [28] (**11a–r**) to obtain the corresponding substituted thiosemicarbazides [29] (**12a–r**). These were cyclised by using an excess of aqueous KOH to form 5-mercapto-1,2,4-triazoles (**13a–r**) in 80–90% yields. Finally, they were reacted with 3-(2-bromoethyl)-1H-indole (**14**) in the presence of trimethylamine to obtain the target molecules (**15a–r**), **as** shown in Figure 1.

The formation of 1H-indole-3-carbohydrazide (**10**) from ethyl 1H-indole-3-carboxylate (**9**) was established by the presence of two singlets corresponding to –CONH- and –NH-NH_2_ protons in the range of δ 9.19 and 4.34 ppm, respectively. The disappearance of the upfield –NH_2_ singlet at δ 4.34 ppm and the presence of four singlets corresponded to the −NH protons at δ 11.71, 9.98, 9.68, and 9.57 ppm, and the additional aromatic protons/aliphatic protons in ^1^H NMR confirmed the formation of thiosemicarbazides (**12a–r**) from hydrazide **10**. The cyclisation of thiosemicarbazides to 5-mercapto-1,2,4-triazoles (**13a–r**) was confirmed by the disappearance of three singlets (δ 9.98, 9.68, and 9.57 ppm), which corresponded to the -NH protons of thiosemicarbazide and the emergence of a singlet at the δ 13.91 ppm characteristic for the -SH proton in ^1^H NMR (Appendix A). Finally, the 1,2,4-triazolo-linked *bis*-indolyl conjugates (**15a–r**) were formed through the S-alkylation of 5-mercapto-1,2,4-triazoles (**13a–r**) with 3-(2-bromoethyl)-1H-indole (**14**), which was confirmed by the disappearance of the -SH proton at δ 13.91 ppm and the appearance of two triplets in the shielded region in the ranges of δ 3.46–3.43 and 3.17–3.13 ppm. The appearance of additional signals corresponding to the indole moiety in 1H NMR (Appendix A) also confirms the formation of 1,2,4-triazolo-linked *bis*-indolyl conjugates (**15a–r**).

### 2.2. Biology

#### 2.2.1. Cytotoxic Activity

The above synthesised conjugates (**15a–r**) were screened for their cytotoxic activity against a panel of nine human cancer cell lines, which included colorectal adenocarcinoma (HCT-15, HT-29, DLD1, and CaCo_2_), lung adenocarcinoma (A549), breast cancer (MCF-7 and MDA-MB-231), glioblastoma, brain cancer (A172), and teratocarcinoma and testicular cancer (TERA-1). These were screened initially at a single concentration of 10 µM by using a (3,4,5-dimethylthiazol-2-yl)-2,5-diphenyl-tetrazolium bromide (MTT) assay. Compounds exhibiting more than 50% growth inhibition were further screened at different concentrations to calculate their IC_50_ values, which are depicted in Table 1. The IC_50_ values were calculated by a dose–response curve, as shown in the Appendix A, and 5-FU was employed as a standard drug.

This class of conjugates demonstrated more susceptibility against the HT-29, HCT-15, MCF-7, and MDA-MB-23 cell lines. Compounds **15a**, **15b**, **15d**, **15f**, **15h**, **15i**, **15k**, **15l**, **15m**, **15o**, and **15r** showed good to moderate cytotoxic activity with IC_50_ < 5 µM against different cancer cell lines. Among them, compounds **15h**, **15k**, and **15r** were found to be the most active against the MDA-MB-231, HCT-15, and HT-29 human cancer cell lines with IC_50_ values of 1.35 μM, 1.37 μM, and 0.85 μM, respectively. Conjugates **15o** and **15r** showed interesting cytotoxic activity in comparison to the standard 5-FU (IC_50_ = 5.31 μM), with IC_50_ values of 0.85 μM and 2.04 μM, respectively, against the HT-29 colorectal human cancer cell line, as shown in Table 1. Thus, detailed biological studies were taken up for the conjugates **15o** and **15r.**

#### 2.2.2. Structure–Activity Relationship (SAR)

Some SAR aspects were drawn from their cytotoxic activity based on (i) the nature of the *N*-substitution (aromatic/aliphatic) on the 1,2,4-triazolyl ring and (ii) the nature and position of the substituent (electron-donating/withdrawing) that is present in the aromatic ring, as shown in Figure 3. Moreover, amongst the aromatic and aliphatic *N*-substituted 1,2,4-triazole conjugates, aliphatic-substituted compounds demonstrated better activity. The nature and degree of the hydrophobicity of aliphatic-substituted compounds also influenced the activity remarkably. Meanwhile, *N*-propyl (**15o**) and *N*-cyclopropyl (**15r**)-substituted conjugates were the most active compared to *N*-butyl, *N*-ethyl, *N*-isopropyl, *N*-cyclohexyl, *N*-allyl, and *N*-benzyl derivatives. The activity profiles of the aliphatic *N*-substituted compounds are depicted in Figure 3. Conjugates **15o** and **15r** showed excellent activity with IC_50_ values of 2.04 µM and 0.85µM, respectively, against the HT-29 cancer cell line. Compound **15h**, with *N*-cyclohexyl substitution, also exhibited promising activity with an IC_50_ value of 1.35 µM against the MDA-MB-231 cancer cell line, while *N*-ethyl-and *N*-butyl-substituted compounds demonstrated moderate activity.

Similarly, among the *N*-aryl substituted compounds, the nature and site of the substituent on the aromatic ring significantly influenced the biological activity. Among the *N*-aryl-substituted compounds, the molecules with methoxy substitution demonstrated enhanced activity when compared to the other substituents. Conjugate **15f**, with 3,4,5-trimethoxy benzene-substituted-1,2,4-triazole, also demonstrated an interesting profile of anti-proliferative activity with IC_50_ values ranging between 2.64 and 8.1 µM against most of the cell lines tested. The site of the substitution also influenced the activity, as was observed in the case of 2-methoxy substitution (**15k**), which resulted in enhanced activity, followed by 3-methoxy (**15l**). Moderate activity was observed in the conjugate with the 4-methoxy group substitution (**15i**).

Among the halogenated compounds, the influence of the substituent on biological activity was observed as Br > Cl > F. Among all the compounds, it was observed that conjugates **15r** and **15o** showed the most promising cytotoxic activity, and thus, they were selected for detailed biological studies.

#### 2.2.3. Cell Cycle Analysis

Among the tested compounds, conjugates **15r** and **15o** showed promising anti-proliferative activity against the HT-29 human cancer cell line with IC_50_ values of 0.85 μM and 2.04 µM, respectively, compared to the standard drug 5-FU (IC_50_ value of 5.31 μM). The effect of these compounds on the progression of cell cycle analysis on the HT-29 cancer cell line was studied. Their profiles in different cell cycle phases were examined using flow cytometry. The results displayed in Figure 4 show that the cell population increased in the G_0_/G_1_ phase, wherein the **15r** and **15o** were inhibited at 65.495% and 65.191%, respectively, at 1 μM and 2 μM concentrations, thus showing the most prominent effect. These conjugates provoked cell cycle arrest at the G_0_/G_1_ phase, thereby suggesting their ability to bind to DNA in addition to inhibiting replication in the HT-29 cancer cell line.

#### 2.2.4. Mitochondrial Membrane Potential (MMP) Analysis

Conversion in the mitochondrial membrane potential is one of the early and main features of cells undergoing programmed cell death. Therefore, to assess whether **15r** and **15o** were engaged in damaging the integrity of the mitochondrial membrane, MMP analysis of these conjugates was performed using the JC-1 dye by flow cytometry in the HT-29 colon cancer cell line. The cells were treated with concentrations of 1 μM and 2 μM of the test compound, respectively. It was observed that there was a significant increase in the monomer formation, depicting the disruption of the mitochondrial membrane potential and a transition in the polarisation of the mitochondria by these conjugates. In contrast, the control cells had negative and intact MMP, Figure 5. Overall, the results show promising effects on the disruption of the mitochondrial potential, suggesting that the conjugates actively act on the mitochondria and reduce the Δψm loss.

#### 2.2.5. Measurement of Cellular Apoptosis in HT-29 Cells

Targeting apoptosis in cancer treatment has been one of the most promising non-surgical therapeutic strategies in curbing cancer. The loss of apoptosis control was shown to increase cancer cell survival and prolong proliferation, thus inducing tumour progression, angiogenesis stimulation, and cell proliferation deregulation. It was observed that **15r** and **15o** significantly increased the early and late apoptotic populations in HT-29 cells. The percentages of early and late apoptotic cell populations significantly increased upon treatment with these conjugates at doses of 1 μM and 2 μM for 24 h compared to the control group.

#### 2.2.6. Effect on ROS Production

Recently, ROS generation therapies have been a prominent target for treating cancer. Agents that induce oxidative stress by increasing ROS production and inhibiting antioxidant defences have received significant attention. Accumulated ROS have been shown to disrupt redox homeostasis, causing severe damage to the cancer cells. Conjugates **15r** and **15o** upregulated the ROS generation in HT-29 colon cancer cells, ultimately leading to cell death. They also increased mitochondrial ROS, superoxide levels, and total ROS generation, thus showing that it is one the most prominent and effective treatment methods for the treatment of cancer (Figure 6).

#### 2.2.7. Immunofluorescence Analysis of TAB-182 Levels and β-Catenin Levels

Several studies have reported that inhibiting the levels of β-catenin and TAB-182 (tankyrase) could be one of the promising strategies in the treatment of a variety of cancers. Typically, cancer cells have been found to express higher levels of β-catenin and TAB-182, which aids in enormous cell proliferation in cancers. Treatment with **15r** and **15o** at doses of 1 µM and 2 µM was found to effectively reduce the expression levels of these proteins in HT-29 colon cancer cells. Further, the treatment was found to inhibit cancer cell progression by inhibiting the nuclear translocation of β-catenin in HT-29 cells in a dose-dependent manner (Figure 7).

#### 2.2.8. Immunofluorescence Analysis of NF-κB and PI3K-P85

NF-κB is an inflammatory marker that is expressed in high levels in cancer cells. The treatment of these conjugates (**15r** and **15o**) was found to prominently reduce inflammatory markers, such as nuclear translocation and the expression levels of NF-κB, along with decreased expression of PI3K-P85 (phosphoinositide 3-kinase). These results suggest that these new conjugates were effective in the treatment of colon cancer and were found to curb the expression of markers that are responsible for cancer cell growth and proliferation. Similarly, F-actin staining and Phalloidin red staining showed prominent damage to the actin filaments in the HT-29 cells, further confirming the anti-proliferative activity of these conjugates (Figure 8).

#### 2.2.9. Western Blotting

The effects of **15o** and **15r** on the expression levels of PI3K p85, β-actin, AXIN-2, TAB-182, and β-catenin were determined by Western blotting. The phosphorylation levels of the PI3K p85 molecule, TAB-182, and β-catenin were significantly inhibited at concentrations of 1µM and 2µM of these conjugates compared to the control. However, no change was observed in the expression levels of the PI3K p85 molecule, TAB-182. Furthermore, it was found that these conjugates reduced the expression levels of the cell proliferation marker β-catenin pathway, as shown in Figure 9. Hence, compounds **15o** and **15r** inhibited the key markers related to cell proliferation via the β-catenin pathway in colorectal cancer.

#### 2.2.10. Molecular Docking Studies

The immunofluorescence assay of the most potent conjugates suggests that this class of compounds inhibited the expression of tankyrase and PI3K levels. Therefore, it was considered of interest to perform in silico docking studies for these newly synthesised conjugates against tankyrases and phosphoinositide 3-kinase (PI3K) proteins using Schrödinger software (Schrödinger’s, LLC, New York, NY, USA) with PDB ID’s 4OA7and 3L54. The docking scores for this series of compounds are illustrated in Table 2. Conjugates **15o** and **15r** show better docking scores against both of the targets (tankyrase and PI3K) and are in correlation with the results of the anti-proliferative activity. The docking poses of the most potent molecules of **15o** and **15r** were investigated, and the 2D and 3D docking poses are shown in Figure 10 and Appendix A.

In PDB ID 4OA7, both non-covalent hydrophobic and hydrophilic interactions were observed between tankyrase protein and these compounds. Conjugate **15r** demonstrated hydrophilic interactions with TYR 1203 and ASP 1198 amino acid residues through hydrogen bonding with a bond length of 2.3 Å. In addition, **15r** showed Pi–Pi stacking with HID 1201 amino acid residues. It also showed hydrophobic interactions with TYR 1213, LE 1212, ALA 1210, MET 1207, ILE 1204, ALA 1202, PHE 1197, LE 1192, ALA 1191, PHE 1188, SER 1186, GLY 1185, and HID 1184 amino acid residues, in addition to some other interactions with LE 1228 and TYR 1224 and some polar interactions with HID 1184 and SER 1186 in chain C. Similarly, **15o** showed interactions with HID 1201, TYR 1203, and TYR 1224 amino acid residues through Pi–Pi stacking. It also showed hydrophobic interactions with PRO 1187, PHE 1188, ALA 1191, ILE 1192, GLY 1196, PHE 1197, ALA 1202, ILE 1204, and MET 1207 amino acids. Moreover, it showed some other interactions with GLY 1211, LE 1212, TYR 1213, GLY 1227, and ILE 1228 and a few polar interactions with HID 1184 and SER 1186 in chain C (Figure 10).

In PDB ID 3L54, phosphoinositide 3-kinase (PI3K) protein and conjugate **15r** showed linkage with TYR 867 amino acid residues through Pi–Pi stacking. It showed hydrophilic interactions with MET 804, PRO 810, TRP 812, ILE 879, VAL 882, and ALA 885, and some other interactions with LE 963, PHE 961, MET 953, and ILE 831, along with some polar interactions with SER 806, THR 887, ASN 951, and HID 967 in chain A. Similarly, conjugate **15o** showed linkage with LYS 833 and ASP 950 amino acid residues through hydrogen bonding, with a bond length of 2.3 Å, and TYR 867 amino acid residues through Pi–Pi stacking. It showed hydrophilic interactions with LE 831, LE 879, LE 881, VAL 882, and ALA 885, and also showed some other interactions with MET 804, ALA 805, TRP 812, PRO 810, PHE 961, LE 963, and MET 953, along with some polar interactions with SER 806, THR 887, and ASN 951 in chain A (Figure 10).

Conjugates **15o** and **15r** were superimposed on both PI3K and tankyrase target binding groves. The phenyl group of the indole ring, triazole NH, and indole NH showed interactions with the ligand present in the protein shown in Figure 10. The docking scores of **15o** and **15r** were −9.614 kcal/mol and −9.223 kcal/mol, respectively, in tankyrase (PDB ID 4OA7), and −6.833 kcal/mol and −6.196 kcal/mol in PI3K (PDB ID 3L54).

The docking results of these compounds are in correlation with the results of the anti-proliferative activity. Conjugate **15r**, with N-cyclopropyl substitution, demonstrated an enhanced anti-proliferative activity profile compared to conjugate **15o** with *N*-propyl substitution, which can also be explicable by the difference in the orientation of their docking poses, as **15r** exhibited more hydrophilic interactions with the tankyrase protein and more hydrophobic interactions with the PI3K protein because 1,2,4-triazole with *N*-substituted groups are more favourable for fitting at the hydrophobic groves. The superimpositions of the docking poses are shown in Figure 10.

#### 2.2.11. Results of In Silico ADME Studies

Generally, a number of drug conjugates have poor ADME profiles, which contribute to their failure in clinical trials. Therefore, a molecule must have a favourable pharmacokinetic profile and biological activity in order to be deemed a potential therapeutic candidate. There are various in silico tools available for predicting the pharmacokinetics of a molecule. Herein, Lipinski’s “Rule of Five” was used to evaluate the “drug-likeness” metrics. All parameters for the tested molecules were well within the prescribed range. The QikProp results suggest that the identified hits have drug-like physico-chemical properties, as depicted in Table 3.

#### 2.2.12. Molecular Dynamic Simulation Studies

The docking complexes of the most active compound **15r**, with the 4OA7 and 3L54 proteins, were subjected to molecular dynamics for a period of 10 nanoseconds by using Schrödinger’s software. The study explored the possible key interactions between the ligand **15r** with the 4OA7 and 3L54 proteins.

During the simulation, a frame was captured every 10 ps and saved into a trajectory. Overall, around 1000 frames were generated throughout the simulation exercise. The root mean square deviation (RMSD) for the protein and ligand was computed by aligning the structures generated during MD simulation in the trajectory with the initial frame. Figure 11A shows the RMSD for the ligand (**15r**)–protein (4OA7) complex, and it is quite evident that the complex was stable for the whole simulation period. However, slight drifts were observed at 5 ns and 7 ns. Later, it stabilised towards the end of the simulation at 1.5Å. Similarly, Figure 11B shows the RMSD for the **15r**-3L54 complex, wherein a slight drift can be observed at 5–6 ns, with stabilisation at 0.9 Å. The root mean square fluctuation (RMSF) was used to analyse the conformational changes occurring along the protein side chain (Figure 11C and 11D). Flexibility within the ranges of 0.5 to 3.5 Å and 0.5 to 3.7 Å can be interpreted from the RMSF data of the proteins 4OA7 and 3L54, respectively. The protein–ligand interactions (interaction of **15r** with PI3K and tankyrase proteins) were also monitored throughout the simulation study, and an analysis report is depicted in Figure 11E and 11F. In both cases, hydrophilic and hydrophobic interactions were observed between the ligand and target proteins. Compound **15r** showed a strong hydrogen bonding with TYR 1203 of 4A07 and as well as with TYR 867 of 3L54. The binding energy profiles of compound **15r** with 4OA7 and 3L54 were ascribed from various components, such as the RMSD, radius of gyration (rGyr), intramolecular hydrogen bonds (intraHB), molecular surface area (MolSA), solvent accessible surface area (SASA), and polar surface area (PSA), as given by ESI.

## 3. Materials and Methods

### 3.1. General

All of the reagents, chemicals, and antibodies utilised in this study were procured from GLR, Merck (India), and Sigma Aldrich, as analytical reagent (AR) grades. Thin-layer chromatography (TLC) plates made of 0.25 mm silica gel were used to monitor the reactions. In the UV cabinet, UV light was employed for the visualisation spots in the reaction mixture. The ^1^H and ^13^C nuclear magnetic resonance (NMR) spectra were determined using the Bruker NMR (400 MHz and 100 MHz) spectrometer. Chemical shifts are expressed as ppm against the TMS internal reference, and the spectra were interpreted using TopSpin software. Using Agilent mass spectrometry, the mass spectra, including MS, were recorded using ESI-MS, whereas a Bruker ALPHA FT-IR spectrometer (Germany) was used to record the IR spectra. The automated melting point apparatus Buchi labortechnik AG 9230 was used to measure the melting points of all synthesised compounds (Switzerland). All the final conjugates prepared in this paper are new and were validated by spectral data analysis. All of the compounds synthesised were recrystallised in methanol and purified using column chromatography with basic alumina. Similarly, other reagents used for the in vitro studies included 3-(4,5-dimethylthiazol-2-yl)-2,5-diphenyltetrazolium bromide (>10T) reagent (Cat no: M2128), normal goat serum (Cat no: 5425), anti-rabbit IgG (Cat no: 7074S), anti-mouse IgG (Cat no: 7076S).

### 3.2. Chemistry

#### 3.2.1. General Procedure for Synthesis of 1H-indole-3-carbohydrazide (**10**)

To the ethanolic (absolute alcohol, 50 mL) solution of 10 g of ethyl 1H-indole-3-carboxylate (**9**), 30 mL of hydrazine hydrate was added, and the reaction mixture was refluxed for 3 h. After the completion of the reaction, observed by TLC, the reaction mixture was slowly brought to ambient temperature and poured onto crushed ice. The white precipitate that formed was filtered off under vacuum to obtain a pure white solid; 98% yield; 1H NMR (500 MHz, DMSO-d6) δ (ppm): 11.55 (s, 1H), 9.19 (s, 1H), 8.16 (d, *J* = 5.0 Hz, 1H), 7.99 (s, 1H), 7.44 (d, *J* = 5.0 Hz, 1H), 7.18–7.10 (m, 2H), 4.34 (s, 2H).

#### 3.2.2. General Procedure for Synthesis of 2-(1H-indole-3-carbonyl)-*N*-phenylhydrazine-1-carbothioamide (**11a–r**)

Hydrazide **10** (0.5 g), taken in absolute alcohol (5 mL), was charged with 1.2 equivalents of different isothiocyanates (**a–r**) (aliphatic/aromatic), and the reaction mixture was refluxed for 4–6 h. After the completion of the reaction, observed by TLC, the reaction mixture was left to reach room temperature and poured onto crushed ice to form a colourless precipitate, which was filtered, washed with cold alcohol, and then dried under vacuum to yield the corresponding thiosemicarbazides (**11a–r**). The representative NMR for 2-(1H-indole-3-carbonyl)-*N*-(4-methoxyphenyl) hydrazine-1-carbothioamide (**11i**) was confirmed by ^1^H NMR (500 MHz, DMSO–d_6_) δ 11.71 (s, 1H), 9.97 (s, 1H), 9.67 (s, 1H), 9.56 (s, 1H), 8.16 (s, 2H), 7.48 (d, *J* = 10.0 Hz, 1H), 7.33 (d, *J* = 5.0 Hz, 2H), 7.20–7.14 (m, 2H), 6.90 (d, *J* = 5 Hz, 2H), 3.75 (s, 3H).

#### 3.2.3. General Procedure for the Synthesis of 5-(1H-indol-3-yl)-4-phenyl-4H-1,2,4-triazole-3-thiol (**13a–r**)

To the ethanolic solution of 0.1 g of 2-(1H-indole-3-carbonyl)-N-phenyl hydrazine-1-carbothioamide (thiosemicarbazide **11a–r**), 2 N aqueous solution of KOH (10 mL) was added and subjected to reflux for 4–6 h. After the completion of the reaction, observed by TLC, the reaction mixture was brought to ambient temperature and poured onto crushed ice, followed by acidification with 2 N HCl to obtain the solid residue. The solid precipitate was then filtered, washed with excess water, dried under vacuum, and recrystallised in ethanol in order to obtain the corresponding pure 5-(1H-indol-3-yl)-4-benzyl-4H-1,2,4-triazole-3-thiol compounds as a mixture of tautomers (**13a–r**)**.** The representative NMR for 5-(1H-indol-3-yl)-4-benzyl-4H-1,2,4-triazole-3-thiol compounds as a mixture of tautomers (**13i**) was formed by ^1^H NMR (500 MHz, DMSO–d_6_) δ 13.92 (s, 1H), 11.49 (s, 1H), 7.44 (d, *J* = 10.0 Hz, 1H), 7.37 (d, *J* = 5.0 Hz, 2H), 7.23–7.15 (m, 4H), 6.41 (d, *J* = 5 Hz, 1H), 3.87 (s, 3H).

#### 3.2.4. General Procedure for the Synthesis of 3-(5-((2-(1H-indol-3-yl)ethyl)thio)-4-benzyl-4H-1,2,4-triazol-3-yl)-1H-indole (**15a–r**)

To the methanolic solution of (10 mL) 0.1 g of 5-(1H-indol-3-yl)-4-benzyl-4H-1,2,4-triazole-3-thiol (**13a–r**) 1.2 equivalents of 3-(2-bromoethyl)-1H-indole **14** was added along with triethylamine (4 equivalent). The reaction mixture was stirred at ambient temperature for 8–10 h. After the completion of the reaction, observed by TLC, the reaction mixture was quenched with an excess of water (30 mL) and extracted with dichloromethane (3 × 20 mL). The combined organic layers were dried over anhydrous sodium sulphate, filtered, and evaporated under vacuum to obtain the crude product, which was recrystallised in ethyl acetate/hexane to obtain the pure target compounds **15a–r**.

3-(5-((2-(1H-Indol-3-yl)ethyl)thio)-4-benzyl-4H-1,2,4-triazol-3-yl)-1H-indole (**15a**)

Light yellow solid 81% yield; melting point of 294.1 °C; ^1^H NMR (400 MHz, DMSO–d_6_) δ 11.54 (s, 1H), 10.80 (s, 1H), 8.05 (d, *J* = 8.0 Hz, 1H), 7.50 (d, *J* = 4.0 Hz, 1H), 7.47 (s, 1H), 7.38 (d, *J* = 8.0 Hz, 1H), 7.27–7.22 (m, 3H), 7.18–7.06 (m, 4H), 7.01–6.98 (m, 1H), 6.94 (d, *J* = 8 Hz, 2H), 6.92–6.89 (m, 1H), 5.31 (s, 2H), 3.34 (t, *J* = 4.0 Hz, 2H), 3.03 (t, *J* = 4.0 Hz, 2H); ^13^C NMR (100 MHz, DMSO–d_6_) δ 152.1, 150.2, 136.7, 136.3, 136.3, 129.4, 128.1, 127.4, 126.3, 126.1, 125.4, 123.5, 122.9, 121.5, 121.4, 120.8, 118.8, 118.8, 112.8, 112.3, 111.9, 102.2, 47.5, 34.2, 25.9; IR (KBr) v 3425, 3053, 2898, 2328.25 1574, 1452, 1390, 1255, 1085, 1002, 939, 838, 745 cm^−1^; HRMS (ESI); *m*/*z* calcd for C_27_H_23_N_5_S [M+H] 450.1954, found 450.1985.

3-(2-((5-(1H-Indol-3-yl)-4-phenyl-4H-1,2,4-triazol-3-yl)thio) ethyl)-1H-indole (**15b**)

White solid 89% yield; melting point of 287.6 °C; ^1^H NMR (400 MHz, DMSO–d_6_) δ 11.36 (s, 1H), 10.86 (s, 1H), 8.24 (s, 1H), 7.61–7.35 (m, 8H), 7.19–7.00 (m, 5H), 6.49 (s, 1H), 3.45 (s, 2H), 3.15 (s, 2H); ^13^C NMR (100 MHz, DMSO–d_6_) δ 153.9, 152.4, 138.8, 138.2, 137.4, 135.9, 132.8, 132.7, 129.5, 127.9, 127.2, 125.6, 125.1, 123.7, 123.6, 122.9, 120.9, 114.9, 114.4, 114.0, 104.4, 48.3, 35.6; IR (KBr) v 3931, 3867, 3612, 3404, 3147, 3043, 2315, 1630, 1569, 1491, 1441, 1386, 1337, 1212, 1095, 997, 939, 824, 733, 688 cm^−1^; HRMS (ESI); *m*/*z* calcd for C_26_H_21_N_5_S [M+H] 436.1591, found 436.1605.

3-(5-((2-(1H-Indol-3-yl)ethyl)thio)-4-(3-bromophenyl)-4H-1,2,4-triazol-3-yl)-1H-indole (**15c**)

Whitish yellow solid 89% yield; melting point of 298.7 °C; ^1^H NMR (400 MHz, DMSO–d_6_) δ 11.38 (s, 1H), 10.87 (s, 1H), 8.23 (d, *J* = 8.0 Hz, 1H), 8.84 (d, *J* = 4.0 Hz, 2H), 7.61–7.54 (m, 3H), 7.45 (d, *J* = 8.0 Hz, 1H), 7.35 (d, *J* = 8.0 Hz, 1H), 7.20–7.10 (m, 3H), 7.09 (d, *J* = 8 Hz, 1H), 7.01 (d, *J* = 8.0 Hz, 1H), 6.60 (s, 1H), 3.44 (t, *J* = 8.0 Hz, 2H), 3.15 (t, *J* = 8.0 Hz, 2H); ^13^C NMR (100 MHz, DMSO–d_6_) δ 151.9, 150.3, 136.8, 136.1, 135.4, 133.8, 130.8, 130.6, 127.5, 125.9, 125.1, 123.6, 123.0, 121.7, 121.6, 120.9, 118.9, 112.9, 112.4, 111.9, 102.4, 46.3, 33.6; IR (KBr) v 3783, 3620, 3421, 3306, 3149, 2850, 2336, 1575, 1439, 1383, 1340, 1250, 1208, 1137, 1092, 1009, 935, 874, 736, 696 cm^−1^; HRMS (ESI); *m*/*z* calcd for C_26_H_20_BrN_5_S [M+H] 514.0705, found 514.0712.

3-(2-((5-(1H-Indol-3-yl)-4-isopropyl-4H-1,2,4-triazol-3-yl)thio)ethyl)-1H-indole (**15d**)

White solid 86% yield; melting point of 256.4 °C; ^1^H NMR (400 MHz, DMSO–d_6_) δ 11.70 (s, 1H), 10.89 (s, 1H), 8.04 (s, 1H), 7.85 (d, *J* = 4.0 Hz, 1H), 7.61 (d, *J* = 8.0 Hz, 1H), 7.51 (d, *J* = 8.0 Hz, 1H), 7.36 (d, *J* = 8.0 Hz, 1H), 7.24–7.20 (m, 2H), 7.14 (t, *J* = 8 Hz, 1H), 7.08 (t, *J* = 8 Hz, 1H), 6.99 (t, *J* = 8 Hz, 1H), 4.02 (t, *J* = 8 Hz, 2H), 3.77 (s, *J* = 8.0 Hz, 1H), 3.52 (t, *J* = 8.0 Hz, 2H), 1.74 (d, *J* = 4.0 Hz, 6H); ^13^C NMR (100 MHz, DMSO–d_6_) δ 153.5, 151.7, 138.7, 138.4, 129.4, 128.3, 127.5, 125.5, 124.8, 123.5, 123.2, 122.7, 120.8, 114.8, 114.3, 113.9, 104.4, 47.9, 36.1, 28.0, 13.2; IR (KBr) v 3639, 3398, 3171, 2924, 1581, 1433, 1345, 1286, 1229, 1116, 1020, 952, 891, 812, 740 cm^−1^; HRMS (ESI); *m*/*z* calcd for C_23_H_23_N_5_S [M+H] 402.1747, found 402.1754.

3-(2-((5-(1H-Indol-3-yl)-4-(p-tolyl)-4H-1,2,4-triazol-3-yl)thio)ethyl)-1H-indole (**15e**)

Pale yellow solid 73% yield; melting point of 278.5 °C; ^1^H NMR (400 MHz, DMSO–d_6_) δ 11.36 (s, 1H), 10.87 (s, 1H), 8.25 (s, 1H), 7.59 (t, *J* = 4.0 Hz, 1H), 7.42–7.40 (m, 2H), 7.35–7.33 (m, 2H), 7.21–7.16 (m, 4H), 7.11–7.04 (m, 2H), 6.99 (t, *J* = 8.0 Hz, 1H), 6.51 (d, *J* = 4.0 Hz, 1H), 3.44 (t, *J* = 8.0 Hz, 2H), 3.14 (t, *J* = 8.0, 2H), 1.19 (s, 3H); ^13^C NMR (100 MHz, DMSO–d_6_) δ 152.8, 150.7, 137.6, 137.3, 134.0, 128.3, 127.1, 126.4, 124.4, 123.7, 122.4, 122.3, 121.6, 119.8, 119.7, 117.9, 113.7, 113.2, 112.8, 103.0, 50.0, 35.1, 26.9; IR (KBr) v 3419, 3047, 2974, 2744, 2673, 2485, 1912, 1619, 1574, 1503, 1444, 1390, 1343, 1234, 1098, 1003, 937, 886, 793, 742 cm^−1^; HRMS (ESI); *m*/*z* calcd for C_27_H_23_N_5_S [M+H] 450.1754, found 450.1761.

3-(2-((5-(1H-Indol-3-yl)-4-(3,4,5-trimethoxyphenyl)-4H-1,2,4-triazol-3-yl)thio)ethyl)-1H-indole (**15f**)

White solid 86% yield; melting point of 322.7 °C; ^1^H NMR (400 MHz, DMSO–d_6_) δ 11.32 (s, 1H), 10.86 (s, 1H), 8.30 (d, *J* = 8.0 Hz, 1H), 7.59 (d, *J* = 8.0 Hz, 1H), 7.43 (d, *J* = 8.0 Hz, 1H), 7.34 (d, *J* = 8.0 Hz, 1H), 7.21–7.13 (m, 3H), 7.07 (t, *J* = 8 Hz, 1H), 6.98 (t, *J* = 8.0 Hz, 1H), 6.86 (s, 2H), 7.67 (d, *J* = 4.0 Hz, 1H), 3.76 (s, 3H), 3.71 (s, 6H), 3.45 (t, *J* = 8.0 Hz, 2H), 3.17 (t, *J* = 8.0 Hz, 2H); ^13^C NMR (100 MHz, DMSO–d_6_) δ 154.1, 152.0, 150.4, 139.0, 136.7, 136.1, 130.3, 127.5, 125.9, 124.8, 123.5, 122.8, 121.9, 121.5, 120.8, 118.8, 118.8, 113.0, 112.2, 111.9, 106.4, 102.6, 62.5, 60.8, 56.9, 46.2, 33.4; IR (KBr) v 3318, 3231, 3059, 2934, 2882, 2837, 2677, 1590, 1502, 1450, 1352, 1305, 1228, 1176, 1125, 989, 944, 848, 802, 733, 700, 646 cm^−1^; HRMS (ESI); *m*/*z* calcd for C_29_H_27_N_5_O_3_S [M+H] 526.1913, found 526.1924.

3-(5-((2-(1H-Indol-3-yl)ethyl)thio)-4-(4-chlorophenyl)-4H-1,2,4-triazol-3-yl)-1H-indole (**15g**)

Pale yellow solid 69% yield; melting point of 289.9 °C; ^1^H NMR (400 MHz, DMSO–d_6_) δ 11.37 (s, 1H), 10.86 (s, 1H), 8.19 (d, *J* = 8.0 Hz, 1H), 7.66 (d, *J* = 8.0 Hz, 2H), 7.56 (dd, *J* = 8.0, 8.0 Hz, 3H), 7.43 (d, *J* = 8.0 Hz, 1H), 7.35 (d, *J* = 8.0 Hz, 1H), 7.21–7.13 (m, 3H), 7.08 (t, *J* = 8.0 Hz, 1H), 6.99 (t, *J* = 8.0 Hz, 1H), 6.62 (d, *J* = 4.0 Hz, 1H), 3.45 (t, *J* = 8.0 Hz, 2H), 3.15 (t, *J* = 8.0 Hz, 2H); ^13^C NMR (100 MHz, DMSO–d_6_) δ 153.9, 152.4, 138.8, 138.2, 137.4. 135.9, 132.8. 132.7, 129.5, 127.9, 127.2, 125.6, 125.1, 123.7, 123.6, 122.9, 120.9, 114.9, 114.4, 114.0, 104.4, 48.3, 35.6; IR (KBr) v 3832, 3682, 3426, 3140, 3053, 2892, 2319, 1575, 1486, 1452, 1390, 1344, 1255, 1212, 1140, 1087, 1002, 938, 837, 747 cm^−1^; HRMS (ESI); *m*/*z* calcd for C_26_H_20_ClN_5_S [M+H] 470.1198, found 470.1207.

3-(5-((2-(1H-Indol-3-yl)ethyl)thio)-4-cyclohexyl-4H-1,2,4-triazol-3-yl)-1H-indole (**15h**)

Yellow solid 72% yield; melting point of 260.6 °C; ^1^H NMR (400 MHz, DMSO–d_6_) δ 11.71 (s, 1H), 10.89 (s, 1H), 7.69 (s, 1H), 7.64 (d, *J* = 8.0 Hz, 2H), 7.52 (d, *J* = 8.0 Hz, 1H), 7.37 (d, *J* = 8.0 Hz, 1H), 7.25–7.19 (m, 2H), 7.14–7.07 (m, 2H), 7.00 (t, *J* = 8.0 Hz, 1H), 4.15 (t, *J* = 12.0 Hz, 1H), 3.59 (t, *J* = 8.0 Hz, 2H), 3.21 (t, *J* = 8.0 Hz, 2H), 2.09 (q, *J* = 12.0 Hz, 2H), 1.81 (d, *J* = 12.0 Hz, 2H), 1.74 (d, *J* = 12.0 Hz, 2H), 1.09 (m, 4H); ^13^C NMR (100 MHz, DMSO–d_6_) δ 151.4, 148.8, 136.7, 136.4, 127.5, 127.0, 126.9, 123.5, 122.6, 121.5, 120.7, 119.8, 118.9, 118.8, 113.0, 112.5, 111.9, 102.2, 65.4, 56.4, 34.1, 31.3, 25.8, 25.0; IR (KBr) v 3427, 3230, 3055, 2928, 2854, 2355, 1581, 1436, 1339, 1298, 1248, 1208, 1105, 998, 947, 886, 817, 731 cm^−1^; HRMS (ESI); *m*/*z* calcd for C_26_H_27_N_5_S [M+H] 442.2054, found 442.2069.

3-(2-((5-(1H-Indol-3-yl)-4-(4-methoxyphenyl)-4H-1,2,4-triazol-3-yl)thio)ethyl)-1H-indole (**15i**)

White solid 83% yield; melting point of 304.7 °C; ^1^H NMR (400 MHz, DMSO–d_6_) δ 11.32 (s, 1H), 10.86 (s, 1H), 8.27 (d, *J* = 8.0, 1H), 7.61 (d, *J* = 8.0, 1H), 7.43–7.34 (m, 4H), 7.21–7.06 (m, 6H), 7.00 (d, *J* = 8.0 Hz, 1H), 6.51 (d, *J* = 4.0 Hz, 1H), 3.85 (s, 3H), 3.34 (t, *J* = 4.0 Hz, 2H), 3.03 (t, *J* = 4.0 Hz, 2H); ^13^C NMR (100 MHz, DMSO–d_6_) δ 160.7, 152.2, 150.7, 136.7, 136.0, 129.9, 127.4, 127.2, 125.8, 124.7, 123.5, 122.9, 121.8, 121.5, 120.8, 118.9, 118.8, 115.7, 112.9, 112.2, 111.9, 102.6, 56.0, 46.2, 33.2; IR (KBr) v 3939, 3767, 3416, 3054, 2836, 1573, 1508, 1450, 1393, 1342, 1301, 1249, 1168, 1015, 939, 833, 741 cm^−1^; HRMS (ESI); *m*/*z* calcd for C_27_H_23_N_5_OS [M+H] 466.1704, found 466.1794.

3-(5-((2-(1H-Indol-3-yl)ethyl)thio)-4-(2-fluorophenyl)-4H-1,2,4-triazol-3-yl)-1H-indole (**15j**)

White solid 85% yield; melting point of 295.8 °C; ^1^H NMR (400 MHz, DMSO–d_6_) δ 11.46 (s, 1H), 10.92 (s, 1H), 8.29 (d, *J* = 8.0 Hz, 1H), 7.79–7.75 (m, 2H), 7.63 (dd, *J* = 8.0, 4.0 Hz, 2H), 7.54–7.48 (m, 2H), 7.39 (d, *J* = 8.0 Hz, 1H), 7.28–7.20 (m, 3H), 7.13 (t, *J* = 8.0 Hz, 1H), 7.04 (t, *J* = 8 Hz, 1H), 6.64 (d, *J* = 4.0 Hz, 1H), 3.49 (t, *J* = 8.0 Hz, 2H), 3.18 (t, *J* = 8.0 Hz, 2H); ^13^C NMR (100 MHz, DMSO–d_6_) δ 151.5, 149.9, 136.4, 135.8, 135.0, 133.5, 130.4, 130.3, 127.1, 125.5, 124.8, 123.2, 122.6, 121.3, 121.2, 120.5, 118.5, 112.5, 112.0, 111.6, 102.0, 45.9, 33.2; IR (KBr) v 3394, 3302, 3159, 3057, 2974, 2924, 2866, 1574, 1504, 1444, 1390, 1345, 1257, 1212, 1143, 1093, 1005, 937, 814, 732, 670 cm^−1^; HRMS (ESI); *m*/*z* calcd for C_26_H_20_FN_5_S [M+H] 454.2091, found 454.2099.

3-(2-((5-(1H-Indol-3-yl)-4-(2-methoxyphenyl)-4H-1,2,4-triazol-3-yl)thio)ethyl)-1H-indole (**15k**)

White solid 89% yield; melting point of 311.2 °C; ^1^H NMR (400 MHz, DMSO–d_6_) δ 11.32 (s, 1H), 10.86 (s, 1H), 8.31 (d, *J* = 4.0 Hz, 1H), 7.64–7.59 (m, 2H), 7.44 (t, *J* = 8.0 Hz, 2H), 7.35 (t, *J* = 8.0 Hz, 2H), 7.22–7.14 (m, 4H), 7.09 (d, *J* = 8.0 Hz, 1H), 7.01 (t, *J* = 8.0 Hz, 1H), 6.51 (s, 1H), 3.69 (s, 3H), 3.39 (t, *J* = 8.0 Hz, 2H), 3.13 (t, *J* = 8.0 Hz, 2H); ^13^C NMR (100 MHz, DMSO–d_6_) δ 159.4, 156.1, 154.4, 140.7, 140.0, 136.6, 134.1, 131.4, 129.8, 127.8, 127.4, 127.2, 126.9, 125.9, 125.8, 125.5, 124.8, 122.8, 117.7, 116.9, 116.2, 115.9, 106.8, 60.4, 37.4, 29.9; IR (KBr) v 3942, 3834, 3683, 3425, 3053, 2973, 2896, 2329, 1913, 1573, 1454, 1388, 1255, 1085, 1002, 938, 837, 745 cm^−1^; HRMS (ESI); *m*/*z* calcd for C_27_H_23_N_5_OS [M+H] 466.1704, found 466.1715.

3-(2-((5-(1H-Indol-3-yl)-4-(3-methoxyphenyl)-4H-1,2,4-triazol-3-yl)thio)ethyl)-1H-indole (**15l**)

White solid 81% yield; melting point of 303.6 °C; ^1^H NMR (400 MHz, DMSO–d_6_) δ 11.34 (s, 1H), 10.86 (s, 1H), 8.26 (d, *J* = 8.0 Hz, 1H), 7.61 (d, *J* = 8.0 Hz, 1H), 7.52 (t, *J* = 8.0 Hz, 1H), 7.43 (d, *J* = 8.0 Hz, 1H), 7.35 (d, *J* = 8.0 Hz, 1H), 7.22–7.14 (m, 4H), 7.11 (d, *J* = 4.0 Hz, 1H), 7.08–6.98 (m, 3H), 6.59 (d, *J* = 4.0 Hz, 1H), 3.77 (s, 3H), 3.45 (t, *J* = 8.0 Hz, 2H), 3.16 (t, *J* = 8.0 Hz, 2H); ^13^C NMR (100 MHz, DMSO–d_6_) δ 160.8, 151.8, 150.2, 136.7, 136.0, 136.0, 131.4, 127.4, 125.8, 124.8, 123.5, 122.9, 121.8, 121.5, 120.8, 120.6, 118.8, 116.4, 114.2, 112.9, 112.3, 111.9, 102.5, 56.1, 33.4, 25.8; IR (KBr) v 3861, 3645, 3561, 3380, 3169, 2921, 1583, 1441, 1337, 1275, 1227, 1096, 1014, 943, 848, 743, 694 cm^−1^; HRMS (ESI); *m*/*z* calcd for C_27_H_23_N_5_OS [M+H] 466.1704, found 466.1707.

3-(5-((2-(1H-Indol-3-yl)ethyl)thio)-4-butyl-4H-1,2,4-triazol-3-yl)-1H-indole (**15m**)

White solid 93% yield; melting point of 269.7 °C; ^1^H NMR (400 MHz, DMSO–d_6_) δ 11.69 (s, 1H), 10.89 (s, 1H), 8.02 (d, *J* = 8.0 Hz, 1H), 7.85 (s, 1H), 7.61 (d, *J* = 8.0 Hz, 1H), 7.51 (d, *J* = 8.0 Hz, 1H), 7.36 (d, *J* = 8.0 Hz, 1H), 7.24–7.13 (m, 3H), 7.08 (t, *J* = 8.0 Hz, 1H), 6.99 (t, *J* = 8.0 Hz, 1H), 4.05 (s, 2H), 3.52 (t, *J* = 8.0 Hz, 2H), 3.18 (t, *J* = 8.0 Hz, 2H), 1.57 (s, 2H), 1.20 (s, 2H), 0.77 (t, *J* = 8.0 Hz, 3H); ^13^C NMR (100 MHz, DMSO–d_6_) δ 151.5, 149.6, 136.7, 136.4, 127.4, 126.3, 125.5, 123.5, 122.8, 121.5, 121.2, 120.7, 118.8, 112.8, 112.3, 111.9, 102.4, 44.2, 34.0, 31.5, 26.0, 19.6, 13.8; IR (KBr) v 3839, 3779, 3708, 3601, 3533, 3459, 3389, 3111, 2937, 2667, 2270, 1919, 1850, 1570, 1434, 1344 1200, 1090, 935, 731 cm^−1^; HRMS (ESI); *m*/*z* calcd for C_24_H_25_N_5_S [M+H] 416.1807, found 416.1915.

3-(5-((2-(1H-Indol-3-yl)ethyl)thio)-4-ethyl-4H-1,2,4-triazol-3-yl)-1H-indole (**15n**)

White solid 93% yield; melting point of 249.4 °C; ^1^H NMR (400 MHz, DMSO–d_6_) δ 11.72 (s, 1H), 10.89 (s, 1H), 8.06 (d, *J* = 8.0 Hz, 1H), 8.87 (d, *J* = 4.0 Hz, 1H), 7.61 (d, *J* = 8.0 Hz, 1H), 7.51 (d, *J* = 8.0 Hz, 1H), 7.36 (d, *J* = 8.0 Hz, 1H), 7.24–7.20 (m, 2H), 7.15 (t, *J* = 8.0 Hz, 1H), 7.08 (t, *J* = 8.0 Hz, 1H), 6.99 (t, *J* = 8.0 Hz, 1H), 4.10 (q, *J* = 8.0 Hz, 2H), 3.52 (t, *J* = 8.0 Hz, 2H), 3.18 (t, *J* = 8.0 Hz, 2H), 1.24 (t, *J* = 8.0 Hz, 3H); ^13^C NMR (100 MHz, DMSO–d_6_) δ 151.3, 149.2, 136.7, 136.5, 127.4, 126.3, 125.4, 123.5, 122.8, 121.5, 121.3, 120.7, 118.9, 118.8, 112.8, 112.3, 111.9, 102.3, 40.6, 34.0, 26.0, 15.4; IR (KBr) v 3892, 3808. 3722, 3596, 3532, 3140, 2902, 2335, 1570, 1440, 1389, 1336, 1231, 1127, 1005, 944, 795, 746 cm^−1^; HRMS (ESI); *m*/*z* calcd for C_22_H_21_N_5_S [M+H] 388.1589, found 388.1596.

3-(2-((5-(1H-Indol-3-yl)-4-propyl-4H-1,2,4-triazol-3-yl)thio)ethyl)-1H-indole (**15o**)

White solid 91% yield; melting point of 260.6 °C; ^1^H NMR (400 MHz, DMSO–d_6_) δ 11.69 (s, 1H), 10.89 (s, 1H), 8.03 (d, *J* = 8.0 Hz, 1H), 7.85 (d, *J* = 4.0 Hz, 1H), 7.61 (d, *J* = 8.0 Hz, 1H), 7.50 (d, *J* = 8.0 Hz, 1H), 7.36 (d, *J* = 8.0 Hz, 1H), 7.24–7.20 (m, 2H), 7.14 (t, *J* = 8.0 Hz, 1H), 7.07 (t, *J* = 8.0 Hz, 1H), 6.99 (t, *J* = 8.0 Hz, 1H), 4.02 (t, *J* = 8.0 Hz, 2H), 3.52 (t, *J* = 8.0 Hz, 2H), 3.17 (t, *J* = 8.0 Hz, 2H), 1.62 (s, *J* = 8.0 Hz, 2H), 0.77 (t, *J* = 8.0 Hz, 3H); ^13^C NMR (100 MHz, DMSO–d_6_) δ 153.5, 151.7, 138.7, 138.4, 129.4, 128.3, 127.5, 125.5, 124.8, 123.5, 123.2, 122.7, 120.8, 114.8, 114.3, 113.9, 104.4, 47.9, 36.1, 28.0, 24.9, 13.2; IR (KBr) v 3884, 3715, 3505, 3404, 3220, 3039, 2966, 2904, 2851, 2675, 1570, 1443, 1389, 1341, 1214, 1115, 1007, 942, 888, 799, 743 cm^−1^; HRMS (ESI); *m*/*z* calcd for C_23_H_23_N_5_S [M+H] 402.1747, found 402.1756.

3-(5-((2-(1H-Indol-3-yl)ethyl)thio)-4-(3-chlorophenyl)-4H-1,2,4-triazol-3-yl)-1H-indole (**15p**)

White solid 89% yield; melting point of 298.1 °C; ^1^H NMR (400 MHz, DMSO–d_6_) δ 11.36 (s, 1H), 10.86 (s, 1H), 8.20 (d, *J* = 8.0 Hz, 1H), 7.66 (d, *J* = 8.0 Hz, 2H), 7.60 (d, *J* = 8.0 Hz, 1H), 7.54 (d, *J* = 8.0 Hz, 2H), 7.43 (d, *J* = 8.0 Hz, 1H), 7.35 (d, *J* = 8.0 Hz, 1H), 7.21–7.13 (m, 3H), 7.08 (t, *J* = 8.0 Hz, 1H), 6.99 (t, *J* = 8.0 Hz, 1H), 6.62 (d, *J* = 4.0 Hz, 1H), 3.45 (t, *J* = 8.0 Hz, 2H), 3.15 (t, *J* = 8.0 Hz, 2H); ^13^C NMR (100 MHz, DMSO–d_6_) δ 151.8, 150.2, 136.7, 136.1, 135.3, 133.8, 130.7, 130.6, 127.4, 125.8, 125.1, 123.5, 123.0, 121.6, 121.5, 120.8, 118.8, 112.8, 112.3, 111.9, 102.3, 46.2, 33.5, 25.8; IR (KBr) v 3563, 3305, 3162, 3062, 2923, 2671, 2487, 2338, 1577, 1438, 1387, 1341, 1248, 1137, 1091, 1016, 938, 875, 737, 687 cm^−1^; HRMS (ESI); *m*/*z* calcd for C_26_H_20_ClN_5_S [M+H] 470.1198, found 470.1261.

3-(5-((2-(1H-Indol-3-yl)ethyl)thio)-4-allyl-4H-1,2,4-triazol-3-yl)-1H-indole (**15q**)

White sticky 68% yield; melting point of 285.3 °C; ^1^H NMR (400 MHz, DMSO–d_6_) δ 11.68 (s, 1H), 10.87 (s, 1H), 8.11 (d, *J* = 4.0, 1H), 7.73 (s, 1H), 7.59 (d, *J* = 8.0 Hz, 1H), 7.49 (d, *J* = 8.0 Hz, 1H), 7.35 (d, *J* = 8.0 Hz, 1H), 7.23–7.15 (m, 3H), 7.09–6.98 (m, 2H), 6.01 (s, 1H), 5.20 (s, 1H), 4.74 (s, 2H), 4.10 (s, 1H), 3.47 (s, 2H), 3.17 (s, 2H); ^13^C NMR (100 MHz, DMSO–d_6_) δ 152.9, 150.8, 137.7, 137.4, 134.1, 128.4, 127.2, 126.5, 124.5, 123.8, 122.5, 122.4, 121.7, 119.8, 119.8, 118.0, 113.8, 113.3, 112.9, 103.1, 50.1, 35.2, 27.0; IR (KBr) v 3821, 3637, 3164, 3047, 2917, 2826, 2356, 1579, 1490, 1430, 1341, 1285, 1233, 1108, 1032, 938, 797, 744 cm^−1^; HRMS (ESI); *m*/*z* calcd for C_23_H_21_N_5_S [M+H] 400.1579, found 400.1596.

3-(5-((2-(1H-Indol-3-yl)ethyl)thio)-4-cyclopropyl-4H-1,2,4-triazol-3-yl)-1H-indole (**15r**)

Creamy white solid 89% yield; melting point of 270.5 °C; ^1^H NMR (400 MHz, DMSO–d_6_) δ 11.68 (s, 1H), 10.88 (s, 1H), 8.17 (d, *J* = 8.0 Hz, 1H), 8.00 (d, *J* = 4.0 Hz, 1H), 7.64 (d, *J* = 8.0 Hz, 1H), 7.49 (d, *J* = 8.0 Hz, 1H), 7.36 (d, *J* = 8.0 Hz, 1H), 7.25 (d, *J* = 4.0 Hz, 1H), 7.21 (t, *J* = 8.0 Hz, 1H), 7.14 (t, *J* = 8.0 Hz, 1H), 7.08 (t, *J* = 8.0 Hz, 1H), 7.00 (t, *J* = 8.0 Hz, 1H), 3.57 (t, *J* = 8.0 Hz, 2H), 3.41 (p, *J* = 4.0 Hz, 1H), 3.21 (t, *J* = 8.0 Hz, 2H), 1.19–1.11 (m, 4H); ^13^C NMR (100 MHz, DMSO–d_6_) δ 153.0, 151.5, 136.7, 136.2, 127.5, 127.0, 126.0, 123.5, 122.6, 121.6, 121.5, 120.6, 118.9, 118.8, 113.1, 112.2, 111.9, 102.7, 46.2, 32.7, 25.6, 9.6; IR (KBr) v 3464, 2860, 2747, 2675, 1578, 1448, 1403, 1342, 1247, 1155, 1093, 1036, 940, 804, 742, 664 cm^−1^; HRMS (ESI); *m*/*z* calcd for C_23_H_21_N_5_S [M+H] 400.1579, found 400.1595.

### 3.3. Biology

#### 3.3.1. MTT Assay

The effects of various 1,2,4-triazolo-linked *bis-indolyl* conjugates (**15a–r**) were evaluated for their anti-proliferative activity using a (3–4, 5-dimethylthiazol-2-yl)-2, 5-diphenyltetrazolium bromide (>10 T) assay. The assay is based on the conversion of tetrazolium-soluble salt into formazan crystals by the mitochondrial enzymes NAD and NADH–dehydrogenase. Colorectal adenocarcinoma (HCT-15 (ATCC-CCL-225™), HT-29 (ATCC-HTB-38™), DLD1 (ATCC-CCL-221™), and CaCo_2_ (ATCC-HTB-37™)), lung adenocarcinoma (A549-ATCC and CCL-185™), breast cancer (MCF-7, ATCC-HTB-22™, MDA-MB-231, and ATCC-HTB-26™), glioblastoma, brain cancer (A172 and ATCC-CRL-1620™), teratocarcinoma, testicular cancer (TERA-1 and ATCC-HTB-105™), and normal rat kidney cells (NRK-52-E and ATCC-CRL-1571™) were plated at a density of 10,000 cells/well in a 96-well plate [30,31]. After the cells attained normal morphology, they were treated with the above conjugates at a single concentration of 10 µM for 24 h. The plate was placed in a humidified CO_2_ incubator at 37 °C. Then, the supernatant was removed, and a >10T solution at a concentration of 0.5 mg/mL was added, followed by incubation at 37 °C for 4 h in a humidified atmosphere. Subsequently, the media was removed, DMSO was added to dissolve the formazan crystals, and then the absorbance was recorded at 570 nm on an I3x Spectramax molecular device. Compounds showing greater than 50% inhibition in the initial screening were further treated at varying doses of 10, 3.33, 1.11, 0.37, 0.12, and 0.041 µM concentrations. The cytotoxicity was expressed as the concentration of compounds that inhibited 50% cell proliferation (IC_50_).

#### 3.3.2. Cell Cycle Analysis

The effects of **15r** and **15o** on the cell cycle of the HT-29 cell line were evaluated using flow cytometry at **15o** (1 µM) and **15r** (2 µM) concentrations. These cells were seeded in a 6-well plate and kept in an incubator at 37 °C for 24 h, and after the attachment of the cells at full confluence, they were treated with the test compounds. After treatment, the cells were harvested and fixed overnight in 75% ice-cold ethanol at 4 °C. Following a PBS wash, the fixed cells were pelleted and stained with RNase (50 U/mL) and PI (20 μg/mL) solution for 20 min in the dark and at room temperature. Further, the cells were analysed using flow cytometry [32].

#### 3.3.3. Evaluation of Mitochondrial Membrane Potential

The JC-1 staining method was used to study the effects of **15r** and **15o** on the mitochondrial membrane potential (Δψm) of HT-29 cells. It was assessed using JC-1 dye, a specific mitochondrial fluorescent probe. Normally Δψm JC-1 forms aggregates with high red fluorescence intensity, so a loss in the Δψm is indicated by a decrease in the red fluorescence and an increase in green fluorescence due to the shifting of the dye from the aggregate to the monomeric form. JC-1 dye was used at a concentration of 2 µM after the seeding and treatment of HT-29 cells in a 6-well plate following incubation for 30 min in an incubator. The red/green fluorescence ratio serves as an indicator of the Δψm loss [33].

#### 3.3.4. Evaluation of Apoptosis by Annexin V/Propidium Iodide (PI)

Annexin V is a very sensitive dye used to detect cellular apoptosis, while PI detects late apoptotic or necrotic populations that are characterised by the loss of integrity of nuclear and plasma membranes. HT-29 cells were plated in a 6-well plate, and after attaining morphology and the desired cell density, treatment was given for 24 h. Then, the cells were harvested and washed in ice-cold PBS and resuspended in annexin-binding buffer. Annexin-V-FITC and PI were used to stain the cells for 5–15 min following cell analysis by flow cytometry [34].

#### 3.3.5. Evaluation of Total Reactive Oxygen Species (ROS)

HT-29 cells were plated in a 6-well plate, and after subsequent treatment, the plate was incubated with CM-H2DCFDA (2′,7′-dichlorodihydrofluorescein diacetate), Sigma Aldrich, at a concentration of 5 µM for 20 min at 37 °C. After the incubation of the dye, the cells were washed with PBS followed by trypsinisation with 0.25% trypsin. Then, the percentage of intracellular ROS generation was measured using flow cytometry (Attune NXT, Thermo Fisher Scientific, Waltham, MA, USA), and 10,000 events were acquired for each treatment group [35].

#### 3.3.6. Evaluation of Mitochondrial ROS

MitoSOX™ Red reagent was used to measure superoxide levels in live cells. MitoSOX™ Red is a novel fluorogenic dye that explicitly targets the mitochondrial membrane, and upon oxidation, produces red fluorescence. HT-29 cells were plated in a 6-well plate and post-treatment-incubated with 5 μM MitoSOX™ Red for 1 h at 37 °C in an incubator. Following incubation, the cells were washed and trypsinised. A total of 10,000 events were run in flow cytometry (Attune NXT, Thermo Fisher Scientific), and the mean fluorescence intensity was measured [36].

#### 3.3.7. Immunocytochemistry (ICC)

HT-29 cells were plated on poly D-lysine-coated coverslips in a 6-well culture plate at a density of 2 × 10^6^ cells per well. Upon treatment with test compounds, the cells were washed with PBS, fixed with 4% paraformaldehyde, and then permeabilised with 0.2% Triton X-100. The cells were blocked with 5% normal goat serum (NGS), washed, and incubated overnight with primary antibodies: β-catenin (1:200 dilution), TAB-182 (1:200 dilution), NF-κB (1:800 dilution), PI3K-P85 (1:200 dilution) at 4 °C overnight incubation. Next, the cells were washed with PBS and then incubated with secondary antibodies, namely, Alexa Flour^TM^ 488 goat anti-rabbit IgG (H+L) ActinGreen and Phalloidin Red, which were added to the cells for 15 min following PBS washing and mounting. Subsequently, the nuclei were stained with Vectashield mounting medium for fluorescence with 4′,6-dia-midino-2-phenylindole (DAPI) (Cat no. H-1200, Vector Laboratories, Burlingame, CA, USA). Negative control slides were prepared by the exclusion of the primary antibody. The slides were kept in a cool place until observation under oil emersion at 63× magnification with a confocal microscope [37].

#### 3.3.8. Western Blotting

HT-29 cells treated with compounds **15o** and **15r** were lysed after 72 h of incubation with 2× SDS lysis buffer containing 0.5 M Tris-HCl, pH 6.8, glycerol, 10% (*w*/*v*) SDS, and a protease inhibitor cocktail. The lysates were sonicated and centrifuged at 16,000× *g* for 20 min, following which the supernatant was collected and subjected to protein estimation using a Pierce TM BCA assay kit (Thermo Fisher). The proteins (30 μg) were separated on SDS gel and transferred to PVDF membranes. The membranes were exposed to blocking buffer containing 5% skim milk and probed with specific primary antibodies at 4 °C overnight. The primary antibodies used were as follows: PI3K p85 (phosphoinositide 3-kinase) (1:1000), β-actin (1:1000), AXIN-2 (1:1000), TAB-182 (tankyrase) (1:1000), and β-catenin (1:1000). Next, the blots were washed three times for 5 min with tris buffer solution with tween-20 (TBST) and incubated with HRP-conjugated secondary antibody for 1 h. The blots were washed three times with 1× TBST, and the bands were detected by ECL Elistar ETA ultra 20 (Cyanagen) [38].

#### 3.3.9. Molecular Docking

All the synthesised compounds were docked against PI3K (PDB ID: 4OA7), and tankyrase (PDB ID: 3L54) protein targets using Schrödinger software. The protein data bank (PDB) was used to obtain the protein structures of the PI3K [39] and tankyrase [40]. (https://www.rcsb.org, accessed on 1 October 2022), with a resolution of 2.301 Å, based on the best resonation, R (free) value, and the number of residues resolved. Initially, Schrödinger’s protein preparation wizard panel was used to analyse the ligand. (Schrödinger’s, LLC, New York, NY, USA). Proteins were present in tetrameric as well as monomeric forms in the workspace. Then, the tetrameric forms were changed into monomeric forms of the protein to later use the OPLS3 2015 force field for protein–energy minimisation in protein preparation. Then, crystallised water molecules were removed and the ligand was retained as such [41]. After protein preparation, a grid was generated around the ligand using receptor grid generation of the glide molecule and applying all the standard glide tool parameters.

Ligand preparation: All eighteen molecules were drawn in ChemDraw, along with the standard, and converted into SDF files. After entering all entities, the ligand preparation command was applied. Ligand docking occurred in this process in the receptor grid, and prepared ligands were subjected to glide XP docking using the standard protocol. After some time, the docking poses and docking scores were obtained and are shown in Figure 10 and Table 2.

#### 3.3.10. ADMET Property Prediction

The QikProp module of Schrödinger software was used to predict the pharmacokinetic properties, such as the molecular weight (˂500 Daltons), predicted octanol/water partition coefficient (QlogP_*o*/*w*_), percentage of human oral absorption, H-bond donors (˂5), and H-bond acceptors (˂10), which were examined for their compliance with Lipinski’s rule of five. The rule explains how molecular characteristics affect the medication pharmacokinetics in the human body, such as absorption, distribution, metabolism, and excretion (ADME). There are various in silico tools available for predicting the pharmacokinetics of a molecule. QikProp module version 5.4 of Maestro was used to calculate the molecular descriptor and predict the ADMET profile of the synthesised compounds [42].

#### 3.3.11. Molecular Dynamic Simulation

Molecular dynamic (MD) simulations were carried out using Schrödinger’s software. MD simulation determines the physical movements of molecules and atoms in the protein–ligand molecular docking complex [43,44]. The MD simulation was performed on the docked complex of compound **15r** with tankyrase1 and PI3K gamma protein (PDB ID: 4OA7 and 3L54) [45,46]. Before performing the MD simulation, the protein was critically vetted for any missing residues using the structure check wizard tool. Ligand energies were minimised using the OPLS3e force field [46]. Later, the ‘Desmond’ program was opened to start the system building, and the protein structure was solvated using the PI3TP water model.

The protein was placed in the centre of the orthorhombic box using a minimised volume, where the distance between any atom of the solute and the edge of the solvent box was at least 10Å. Counter-ions were added to neutralise the system, and the salt ion concentration was set to 0.15M based on the psychological strength. After adding all the parameters, the process was started using a standard protocol for system building (Desmond version). Following the energy minimisation, a Nose–Hoover chain thermostat was used to carry out the NPT equilibration at 310 K to maintain a constant temperature. Additionally, a Martyn–Tobias–Klein barostat was employed to maintain a pressure of 1 bar [47]. MD simulations were carried out with the default settings of the normal calculation method, with the following parameters: the simulation length was 10 ns, and the solvent model was an explicit method. The periodic boundary conditions were taken into consideration while performing the MD simulations to avoid edge effects. The energy of the protein–ligand complex was minimised to 0.25 kcal/mol. After the MD simulation was completed, the trajectory was examined for RMSD and RMSF plots as well as protein–ligand contacts.

## 4. Conclusions

In conclusion, a series of eighteen 1,2,4-triazolo-linked *bis*-indolyl conjugates (**15a–r**) were prepared in a multistep synthetic methodology. All the synthesised compounds were screened for their cytotoxic activity against a panel of nine different human cancer cell lines. Among them, conjugates **15a**, **15b**, **15d**, **15i**, **15l**, **15f**, **15h**, **15k**, and **15m** illustrated good cytotoxic activity against different cancer cell lines. Conjugates **15o** and **15r** displayed the most promising activity with IC_50_ values of 2.04 μM and 0.85 μM, respectively, against colorectal adenocarcinoma (HT-29). These two conjugates were found to induce cell cycle arrest at the G_0_/G_1_ phase and also enhanced cellular ROS production with increased superoxide levels. They demonstrated prominent effects in the destruction of the mitochondrial membrane potential of cancer cells. They also significantly increased the early and late apoptotic cell populations, as evidenced by annexin/PI staining. Immunofluorescence assays in the latter experiments revealed crucial changes in the expression levels of proteins responsible for cancer cell growth and proliferation. Interestingly, **15r** and **15o** significantly reduced the expression levels of TAB-182, PI3K-P85 and also inhibited the nuclear translocation of NF-κB and β-catenin. The Western blot results show that compounds **15o** and **15r** inhibited the expression levels of PI3K p85, β-actin, AXIN-2, TAB-182, and β-catenin proteins, as these markers are involved via the β-catenin pathway in colorectal cancer. The above experiments revealed that these conjugates possess promising cytotoxic activity and have immense potential to become possible leads for the treatment of colon cancer.

The results of the in silico studies show that the conjugates **15o** and **15r** demonstrated different hydrophobic/hydrophilic interactions with the target proteins with good docking scores of −9.614 kcal/mol and −9.223 kcal/mol, respectively, in tankyrase (PDB ID 4OA7), and −6.833kcal/mol and −6.196 kcal/mol in PI3K (PDB ID 3L54). Interestingly, all the synthesised conjugates were within the ADME parameters, and thus, they seem to be druggable in nature. Moreover, the molecular dynamic simulation studies demonstrated that the RMSD value of **15r** and the protein (4OA7 and 3L540) complexes did not exceed 2.0 Å, and the relative mean square fluctuation (RMSF) plot fluctuations of the amino acid residues were less than 5.0 Å, thereby indicating the stability of the protein conformation. This was followed by MM-GBSA calculation to identify the interaction pattern and strength of the interaction. The combination of MD simulation and experimental techniques is an a priori means to solve biological problems and provides an in-depth understanding of the relationship between protein structure and function. Therefore, based on the immunofluorescence assay, Western blot assay, and docking results, it appears that conjugates **15r** and **15o** act as dual inhibitors of tankyrase and PI3K in colorectal cancer. Therefore, this class of conjugates could serve as an excellent template for the discovery and development of tankyrase and PI3K dual inhibitors.

## Data Availability

Data are contained within the article and some data are present in the Appendix A.

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
