# Peer review of "Synthesis and Cytotoxic Activity of 1,2,4-Triazolo-Linked Bis-Indolyl Conjugates as Dual Inhibitors of Tankyrase and PI3K"

_molecules, 2022, doi:10.3390/molecules27217642_

Round 1

Reviewer 1 Report

The reviewed paper is devoted to the synthesis and study of the cytotoxic activity of compounds containing two indole and one 1,2,4-triazole cycles. Eighteen novel compounds were obtained in the paper and their structure was confirmed by IR, NMR and mass spectral data. Screening of the compounds obtained for the cytotoxic activity against a panel of nine different human cancer line was completed. Numerous methods including cell cycle analysis, mitochondrial membrane potential analysis, measurement of cellular apoptosis, immunofluorescence analysis, molecular docking and others, were successfully used. Some of the compounds displayed a promising activity against colorectal adenocarcinoma. Unfortunately, the paper contains few drawbacks.

1.      Synthetic procedure should be written according to the rules of the journal.

2.      Compounds not described in the literature must be fully characterized, including elemental analysis or HRMS, LRMS and IR spectra. For known compounds, a comparison of the literature data (for example, mps) with the data obtained in the paper should be given.

3.      The number of signals in the NMR spectra 1H or/and 13C for many compounds does not coincide with the calculated values.

4.      The paper contains a number of typos that should be corrected.

The paper can be recommended for publication in the journal ‘Molecules’ after major revision.

Author Response

REVIEWER #1

Recommendation: The paper can be recommended for publication in the journal ‘Molecules’ after major revision.
Response: We thank and appreciate the reviewer for the positive response.

Comments 1: Synthetic procedure should be written according to the rules of the journal.

Response: As suggested, synthetic procedure is changed according to the journal’s format.

Comments 2: Compounds not described in the literature must be fully characterized, including elemental analysis or HRMS, LRMS and IR spectra. For known compounds, a comparison of the literature data (for example, mps) with the data obtained in the paper should be given.

Response: HRMS and IR spectra of all the compounds has been included in ESI.

Comments 3: The number of signals in the NMR spectra 1H or/and 13C for many compounds does not coincide with the calculated values.

Response: Corrections were made as suggested.

Comments 4: The paper contains a number of typos that should be corrected.

Response: Modified as suggested.

Reviewer 2 Report

Kamal and co-workers designed and synthesized a new series of  1,2,4-triazolo tethered bis-indolyl conjugates and screened them for their cytotoxic activity against various human cancer cell lines.  Multistep synthesis was carried out to obtained the desired compounds straightforwardly in excellent yields.  Two hit Conjugates, 15o  and 15r illustrated  promising cytotoxicity against against HT-29 cell line. Further biological studies revealed that  they induce apoptosis and ROS production while reducing the expression levels of PI3K-P85, β-catenin, 31 TAB-182, β-actin, AXIN-2 and NF-κB markers. In-silico studies also correlates with the biological assays results.

Overall, the article has high merit and is scientifically sound.

The article is well designed and drafted with some minor corrections.

1. In scheme-1,'Reflux is mistakenly written as 'Refulx'.

2. IC50 value of 5-FU must also be included in Table 2.

3. In ESI, 13C of some compounds (S19, S24 and S27) should be revised with more scans.

Author Response

RESPONSES TO REVIEWER #2

Recommendation: The paper can be recommended for publication in the journal ‘Molecules’ after minor revision.

Response: We thank and appreciate the reviewer for the positive response.

Comments:

Comment 1: In scheme-1,'Reflux is mistakenly written as 'Refulx'.

Response: Refulx is changed to Reflux as suggested.

Comment 2: IC50 value of 5-FU must also be included in Table 2.

Response:  Correction is done as suggested by reviewer.

Comment 3: In ESI, 13C of some compounds (S19, S24 and S27) should be revised with more scans.

Response:  13C-NMR spectra of compounds 15e, 15m and 15q has been improved with a greater number of scans as suggested by the learned reviewer.

Round 2

Reviewer 1 Report

The authors responded to all comments of the reviewer. The paper may be published after the authors include the calculated values for high-resolution mass spectra in the experimental part.

Author Response

REVIEWER #1

Recommendation: The paper can be recommended for publication in the journal ‘Molecules’ after minor revision.
Response: We thank and appreciate the reviewer for the positive response.

Comments 1: The paper may be published after the authors include the calculated values for high-resolution mass spectra in the experimental part.

Response: As suggested, calculated values for high-resolution mass spectra are inserted in experimental part.